# Antibody-Drug Conjugates: The New Treatment Approaches for Ovarian Cancer

**DOI:** 10.3390/cancers16142545

**Published:** 2024-07-15

**Authors:** Sho Sato, Tadahiro Shoji, Ami Jo, Haruka Otsuka, Marina Abe, Shunsuke Tatsuki, Yohei Chiba, Eriko Takatori, Yoshitaka Kaido, Takayuki Nagasawa, Masahiro Kagabu, Tsukasa Baba

**Affiliations:** Department of Obstetrics and Gynecology, Iwate Medical University, Yahaba 028-3694, Iwate, Japan; satos@iwate-med.ac.jp (S.S.); amipc7@gmail.com (A.J.); a_note_to_follow_sew@outlook.jp (H.O.); nmrhappy@gmail.com (M.A.); 412.sailing@gmail.com (S.T.); ychiba@ncc.go.jp (Y.C.); takatori@iwate-med.ac.jp (E.T.); ykaido@iwate-med.ac.jp (Y.K.); tnagasaw@iwate-med.ac.jp (T.N.); mkagabu@iwate-med.ac.jp (M.K.); babatsu@iwate-med.ac.jp (T.B.)

**Keywords:** ovarian cancer, antibody-drug conjugates, platinum-resistant, cancer recurrence

## Abstract

**Simple Summary:**

Antibody-drug conjugates (ADCs) are a promising new treatment modality for patients with cancer. They have been approved by the US Food and Drug Administration for treating breast, gastric, cervical, and ovarian cancers (OC), as well as lymphoma and multiple myeloma. Recently, several ADCs have undergone clinical trials for OC, and their development is underway. Several unmet medical needs exist in OC, including treatment for patients with platinum recurrence. This new treatment modality may benefit these patients. ADCs, a new concept of agents, comprise an antibody, a linker, and a payload. If the target is expressed in tumors, the payload specifically reaches the tumor cells. This approach is particularly suitable for OC because of its heterogeneous nature. In this review, we describe the existing evidence for ADC use in OC treatment and discuss ongoing clinical trials.

**Abstract:**

Ovarian cancer (OC), accounting for approximately 200,000 deaths worldwide annually, is a heterogeneous disease showing major differences in terms of its incidence, tumor behavior, and outcomes across histological subtypes. In OC, primary chemotherapy, paclitaxel carboplatin, bevacizumab, and PARP inhibitors have shown prolonged progression-free survival and a favorable overall response rate compared to conventional treatments. However, treatment options for platinum-resistant recurrence cases are limited, with no effective therapies that significantly prolong the prognosis. Recently, mirvetuximab soravtansine, an alpha-folate receptor (FRα)-targeted antibody-drug conjugate (ADC), was approved by the US Food and Drug Administration for patients with FRα-positive recurrent epithelial OC (EOC). This approval was based on a Phase II study, which demonstrated its efficacy in such patients. ADCs comprise an antibody, a linker, and a payload, representing new concept agents without precedence. Advanced clinical studies are developing ADCs for patients with OC, targeting solid tumors such as gynecologic cancer. Ongoing clinical trials are evaluating ADCs targeting FRα and human epidermal growth factor receptor 2, trophoblast cell surface antigen-2, sodium-dependent phosphate transport protein 2B, and cadherin-6 in Phase II/III studies. In this review, we summarize the existing evidence supporting the use of ADCs in OC, discuss ongoing clinical trials and preclinical studies, and explore the potential of these innovative agents to address the challenges in OC treatment.

## 1. Introduction

Ovarian cancer (OC) accounts for approximately 200,000 deaths worldwide annually and is the fifth leading cause of cancer-associated mortality in women in the USA [1,2]. It is a heterogeneous disease characterized by notable differences in incidence, tumor behavior, and outcomes across various histological subtypes [3].

In OC, primary chemotherapy, paclitaxel carboplatin with or without bevacizumab (Bev) [4,5], and PARP inhibitor (PARPi) maintenance treatment [6,7,8] have prolonged progression-free survival (PFS) and achieved a favorable overall response rate (ORR) compared to conventional treatments. If patients show platinum-sensitive recurrence (PSR), administering platinum combination chemotherapy [9,10,11] with Bev and Bev maintenance [12,13,14] or PARPi maintenance [15,16] provides a longer PFS and better ORR. In the case of platinum-resistant recurrence (PRR), treatments are limited [17,18]. Even with monotherapy, the ORR is only 20–30% [19,20,21,22,23,24,25,26]. However, combining monotherapy with Bev considerably prolongs PFS compared to using monotherapy alone [27]. For patients with PRR, clinical trials of immunotherapy have been attempted; however, the results were not comparable with those of standard therapy [28,29]. In OC, including fallopian tube and peritoneal cancers, treatment for PRR remains a significant challenge. While the development of Bev and PARPi has prolonged PFS, it has not markedly extended overall survival (OS) compared to standard therapy in primary or PSR treatment. Thus, addressing the urgent need for new therapeutics in OC is crucial.

Recently, tisotumab vedotin (TV) was approved for patients with cervical cancer (CC). TV is the first antibody-drug conjugate (ADC) for gynecologic cancers approved by the US Food and Drug Administration (FDA). Mirvetuximab soravtansine (MIRV) was approved by the FDA for patients with alpha-folate receptor (FRα)-positive recurrent epithelial OC (EOC). This was the first ADC to obtain FDA approval for OC. ADCs are a new concept of targeted therapy comprising an antibody, a linker, and a payload. The antibody selectively targets tumor cells, while cytotoxic agents as a payload are used to achieve the treatment effect.

In this review, we discuss the mechanisms underlying the efficacy of ADCs against tumors and examine the existing evidence for their use in treating gynecologic cancers, especially OC. We also discuss ongoing clinical trials and preclinical studies, highlighting the progress in ADC development as a new treatment approach for patients with OC.

## 2. Mechanisms of ADCs

As shown in Figure 1A, the antibodies in ADCs are responsible for target recognition and delivery. The small-molecule drugs as payloads are responsible for drug efficacy. The stability of the linker in vivo ensures that cytotoxic molecules are not launched prematurely and drug cleavage occurs only when it enters the tumor, preventing systemic toxicity [30,31,32].

Antibodies such as immunoglobulin G (IgG) serve as the primary antibody backbone in this class of therapeutics, particularly those commonly used in ADCs. IgG2 and IgG4, with their long circulation half-lives and high affinity, are often used in ADCs. While IgG3 is immunogenic, it is generally avoided due to its short half-life [33]. The linker technology in ADCs has evolved significantly. Linkers serve two key roles. First is stability, that is, ensuring the cytotoxic payload remains attached to the antibody while circulating in plasma. The second is efficient release, that is, facilitating payload release within the tumor. Linkers are cleavable (e.g., pH-sensitive hydrazone) or non-cleavable (e.g., thioether) [33]. The cytotoxic molecules used as payloads in ADCs are limited and can be categorized into microtubule-targeting and DNA-damaging agents; however, the next-generation ADCs involve RNA polymerase inhibitors and other agents [34]. Microtubule-targeting agents are the most commonly used payloads, targeting maytansine (DM1 and DM4) and vinca alkaloid (monomethyl auristatin E (MMAE) and MMAF) sites. They inhibit tubulin, break microtubules, and hold the cell cycle in the G2/M phase, resulting in cell death. However, these toxic events occur only in functional proliferating cells [35], whereas DNA-damaging agents affect both proliferating and non-proliferating cells owing to their independence of the cell cycle [36]. The process of targeting tumors with ADCs is as follows: the antibody in an ADC combines with an antigen on the tumor. The ADC is internalized through receptor-mediated endocytosis and divided into antibodies and payloads through endolysosomal processing. The payload is released into the cytoplasm and causes cell death (Figure 1B).

The difference between ADCs and cytotoxic agents is selective targeting. Cytotoxic agents used in chemotherapy affect cell death through DNA damage or microtubule disruption without selective targeting [33]. For example, achieving the desired effect on tumors with conventional chemotherapeutic agents required controlling the dosage, which often led to significant side effects. In contrast, conventional targeted therapies selectively affect cells but rely on indirect mechanisms such as antibody-dependent and complement-dependent cellular cytotoxicity, which do not provide sufficient direct cytotoxicity. Thus, ADCs have been developed to address these limitations by utilizing their ability to selectively deliver cytotoxic drugs to tumor cells via antigen-antibody interactions [37,38].

## 3. Evidence of ADC efficacy

### 3.1. The Effectiveness of ADCs against Gynecologic Cancers

The effectiveness of ADCs against gynecologic cancers (CC) has been reported in Phase II/III trials (Table 1), and several of these agents have been approved by the FDA. For instance, TV was approved for patients with recurrent or metastatic CC. It consists of an antibody targeting tissue factor (TF), a protease-labile Val-Cit-PABA linker, and MMAE as a payload. Coleman et al. have reported a Phase II clinical trial for TV against recurrent or metastatic CC. The ORR was 24% (24/102), and seven patients showed a complete response (CR). The most common treatment-related adverse events (TRAEs) included alopecia (38%), epistaxis (30%), nausea (27%), conjunctivitis (26%), fatigue (26%), and dry eyes (23%) [39]. In a Phase III trial with OS as the primary endpoint, prolonged survival was observed with TV compared to that with the investigator’s choice of monotherapy using topotecan, vinorelbine, gemcitabine, or irinotecan (9.5 months vs. 11.5 months; hazard ratio (HR): 0.70; 95% CI: 0.54–0.89) [40]. These results were reported during a congress held by the European Society for Medical Oncology (ESMO) in 2023. Trastuzumab deruxtecan (T-DXd) has also been reported to benefit patients with recurrent or metastatic CC expressing human epidermal growth factor receptor 2 (HER2). T-DXd consists of an antibody targeting HER2, a cleavable tetra peptide linker, and deruxtecan (a topoisomerase I inhibitor) as a payload. The ORR was 50.0% (95% CI: 33.8–66.2) [41]. This Phase II trial was conducted as a basket trial for solid tumors, including OC and endometrial cancer (EC).

T-DXd has been evaluated in Phase II trials for patients with recurrent uterine carcinosarcoma (UCS) and EC, respectively. Nishikawa and Hasegawa et al. have reported one such trial for patients with recurrent UCS expressing HER2. The ORR by central review in the HER2-high and -low groups were 54.5% (95% CI: 32.2–75.6) and 70.0% (95% CI: 34.8–93.3). respectively. The most common TRAEs included decreased neutrophil count (27%), anemia (24%), and decreased lymphocyte count (21%) [48]. Simultaneously, in a co-clinical study for T-DXd against HER2-expressing UCS, Yagishita et al. found patient-derived xenograft (PDX) models suitable for demonstrating the efficacy of ADCs [50]. Meric-Bernstam et al. have conducted a Phase II trial of T-DXd for patients with recurrent EC expressing HER2, with an ORR of 57.5% (95% CI: 40.9–73.0) [41]. A Phase II trial of sacituzumab govitecan-hziy (SG) has also been conducted for patients with recurrent EC expressing high levels of trophoblast cell surface antigen-2 (TROP2). SG consists of an antibody targeting TROP2, a cleavable CL2A linker, and SN-38 (an active metabolite of irinotecan) as payload, with an ORR of 35% [49]. 

### 3.2. The Effectiveness of ADCs Efficacy against OC

As shown in Table 1, a few phase II/III trials of ADCs for patients with FRα-positive recurrent OC have been reported. FRα, encoded by the FOLR1 gene, has been focused on due to its high expression in several cancer types including OC, and is highly expressed in approximately 50-80% of patients with OC [51,52]. The FDA-approved MIRV consists of an antibody targeting FRα, a sulfo-SPDB disulfide linker, and DM4 as payload [53]. First, a Phase III trial of MIRV for patients with FRα-expressing platinum-resistant OC was conducted, with the primary end point being PFS. Although MIRV did not significantly prolong PFS compared with the investigator’s choice of monotherapy, secondary endpoints consistently favored MIRV, particularly in patients with high FRα expression [42]. Thus, the Phase II SORAYA trial was conducted for patients with FRα-high platinum-resistant recurrent high-grade serous OC (PRR-HGSOC). The ORR was 32.4% (34/105; 95% CI: 23.6–42.2), and five patients showed CR. The most common TRAEs were blurred vision (41%), keratopathy (29%), and nausea (29%) [45]. A Phase III trial of MIRV for patients with FRα-high PRR-HGSOC has been reported. The primary endpoint was PFS again. MIRV prolonged PFS compared with the investigator’s choice of monotherapy with paclitaxel, pegylated liposomal doxorubicin, or topotecan (5.62 months vs 3.98 months; p < 0.001) [43]. This trial revealed that MIRV showed a differentiated and more manageable safety profile than chemotherapy; other Phase III trials also showed this trend. A Phase II trial of T-DXd was also conducted for patients with HER2-expressing PRR. The ORR was 45.0% (95% CI: 29.3–61.5) [41].

## 4. ADCs as a Promising Treatment for Patients with OC

To determine the current situation of ADC development for treating gynecologic cancer, we list the ongoing Phase II/III trials for OC in Table 2. While the development of ADCs is advancing rapidly, this table presents data as of March 2024.

The development of ADCs targeting FRα, TROP2, and HER2 is based on previous reports on efficacy against gynecologic cancer in Phase II/III trials (Table 1). Reports suggest that FRα, TROP2, and HER2 exhibit higher expression in OC tumor tissues, as verified by IHC, approximately at 50–80% [51,52], 50–60% [54,55], and 12–30% [56,57] respectively. These targets show no or low-level expression in normal tissue.

More clinical trials have been conducted for OC compared with other gynecologic cancers. New targets for ADCs, apart from FRα, TROP2, and HER2 are under development for patients with recurrent OC.

Although only MIRV was approved by the FDA, other FRα-targeted ADCs are also being developed in Phase II/III studies. MORAb-202 consists of an antibody targeting FRα, a cathepsin-B–cleavable linker, and eribulin mesylate as payload. A Phase I study of MORAb-202 was conducted for patients with PRR OC [58,59]. The most common TEAE was interstitial lung disease. Other common TEAEs across all grades in Cohort 1 (0.9 mg/kg) and 2 (1.2 mg/kg) included nausea (25.0%; 33.3%), pyrexia (33.3%; 42.9%), malaise (16.7%; 28.6%), and headache (12.5%; 47.6%); ORR was 25.0% and 52.4%, respectively. Luveltamab tazevibulin (luvelta; STRO-002) is also an ADC that targets FRα. This agent consists of an antibody targeting FRα, SC239 drug-linker, and 3-aminophenyl hemiasterlin as payload [60]. A Phase I study of STRO-002 was conducted for patients with PRR OC [61]. The most common TEAEs included neutropenia (70.5%), arthralgia (18.2%), and anemia (13.6%). However, G3/4 neutropenia had a higher incidence with 5.2mg/kg luvelta than with 4.3 mg/kg (76% vs 65%); the ORR was 31.3% and 43.8%, respectively.

Bignotti et al. have reported that TROP2 overexpression is an independent marker of poor prognosis in OC, promoting increased proliferation, invasion, and metastases [62]. TROP2-targeted ADCs are currently under development; for example, datopotamab deruxtecan (Dato-Dxd), which consists of an antibody targeting TROP2, a cleavable tetra peptide linker, and deruxtecan as payload. In an in-vitro study, Okajima et al. showed that Dato-Dxd exhibits antitumor activity against TROP2- expressing tumors through efficient payload delivery and acceptable safety profiles in preclinical models, including OC [63]. A phase I trial of Dato-Dxd for patients with lung and breast cancer also demonstrated encouraging efficacy and safety [64,65]. Consequently, a phase II study of Dato-Dxd for patients with recurrent OC is underway. Other TROP2-targeted ADCs including SG, which demonstrated efficacy for recurrent EC in a Phase II study [49], are being developed for patients with PRR OC (Table 2). SG has received FDA approval for the treatment of urothelial cancer [66] and breast cancer [67]. 

T-DXd, a HER2-targeted ADC, demonstrated efficacy for patients with PRR OC in a Phase II trial [41]. This trial enrolled several solid tumors, including OC, EC and CC. The FDA will give accelerated approval for T-DXd to treat any advanced solid cancer that expressed HER2 as a result of this trial. Other HER2-targeted ADCs under development include disitamab vedotin (RC48), which consists of an antibody targeting HER2, a protease-cleavable linker, and MMAE as payload [68]. In other solid tumors, particularly in HER2-positive gastric cancer, RC48 showed considerable safety and efficacy in Phase I/II trials [69,70,71]. Trastuzumab rezetecan (SHR-A1811), another HER2-targeted ADC, consists of an antibody targeting HER2, a stable and a cleavable tetrapeptide-based linker, and SHR9265 (a topoisomerase I inhibitor) as payload [72]. In a Phase I trial, SHR-A1811 showed favorable antitumor activity and an acceptable safety profile for patients with heavily pretreated HER2-expressing advanced non-breast solid tumors, including UC [73]. As shown in Table 2, these HER2-targeted ADCs have progressed to Phase II trials for patients with OC. T-DXd and RC48 have been approved by the FDA for breast cancer treatment [74,75]. 

Sodium-dependent phosphate transport protein 2B (NaPi2b) is also a potential target for ADCs [76]. Levan et al. have reported that NaPi2b is expressed in 93% (127/136) of patients with OC [77]. Lifastuzumab vedotin (LIFA) was developed as NaPi2b-targeted ADCs, consisting of NaPi2b as the target, a cleavable maleimidocaproyl-valyl-citrullinyl-p-aminobenzyloxycarbonyl (mc-val-cit-PABC) type linker as the linker, and MMAE as the payload. Phase II study of LIFA was conducted, and the primary end point was PFS. In this Phase II study, it was reported that PRR OC had no significant effect compared with PLD. Toxicities included grade 3 TRAEs (46% LIFA; 51% PLD) and serious TRAEs (30% both arms) [46]. Richardson et al. demonstrated promising single-agent antitumor activity and a favorable tolerability profile of Upifitamab rilsodotin (UpRi), a NaPi2b-targeted ADC, in heavily pretreated patients with PRR OC in a Phase Ib/II trial. UpRi has NaPi2b as the target, a novel scaffold-linker as the linker, and AF-HPA as the payload. The ORR was 34%. The most common TRAEs above grade 3 were a transient increase in AST levels, fatigue, anemia, and thrombocytopenia. No severe TRAEs of neutropenia, peripheral neuropathy, or ocular toxicity have been reported. Less frequent grade 3 TRAEs, including pneumonitis, and lower rates of dose reduction and discontinuation were observed in those treated with a lower dose [78]. UpRi has potential as a maintenance therapy for PSR OC because the majority of HGSOCs are estimated to have high NaPi2b expression [79].

Cadherin-6 (CDH6) is also under focus as a potential target of ADCs for treating OC, as reported by Shintani et al. Approximately 64.6% (117 of 181) of patients with primary EOC tested positive for CDH6, and these patients had shorter PFS and OS than CDH6-negative patients. Moreover, CDH6 expression was observed in 74.5% (38 of 51) of patients with recurrent EOC [80]. Raludotatug deruxtecan (R-Dxd) was developed as a CDH6-targeted ADC. It consists of an antibody targeting CDH6, a protease-cleavable ma leimide Gly-Gly-Phe-Gly tetrapeptide-based linker, and deruxtecan as a payload. Additionally, in an in-vitro study, Shintani et al. demonstrated the inhibition of cell growth by R-DXd using cell models derived from patients with OC that were CDH6-positive [81]. Moore et al. have reported a Phase I trial of R-Dxd for recurrent OC, wherein the most common all-grade TEAEs were nausea (55%), fatigue (40%), vomiting (38%), and diarrhea (33%), and the ORR was 38% (13 of 34) [82]. Thus, in Phase II/III trials, R-Dxd is expected to have better efficacy than conventional monotherapy against PRR OC.

Thus, we discuss ongoing Phase II/III trials of ADCs for OC which are expected efficacy in this section. At the end of 2023, there were 13 FDA-approved ADCs for cancer. Some ADCs, SG, T-DXd, and RC48, which we mentioned in this section, are among them [83]. These ADCs showed comparable TRAEs in clinical trials for other cancers, and these can be managed feasibly compared with other ADCs.

Additionally, ADCs and the development of biomarkers for ADCs in treatment selection and treatment monitoring need to be feasible for OC. Criteria for enrolled patients in clinical trials of ADCs depend on whether the target is expressed using IHC in tumor tissue. Although tumor tissue is useful for selecting treatment, it is not useful for treatment monitoring. Biomarkers such as non-invasive and easy are needed. Kurosaki et al. reported the correlation between serum FRα expression levels and tumor FRα expression in OC, suggesting that serum FRα might be a useful non-invasive serum biomarker for FRα targeted therapy [84]. The development of serum biomarkers in other targets for ADCs is expected. In the near future, treatment selection for ADCs by performing blood tests on each patient is expected.

## 5. Treatment-Related Adverse Events with ADCs

Although ADCs target the antigens expressed on tumors, they can also lead to toxicity. These toxicities not only depend on the payload, but are also influenced by the nature of the linker, the presence of the target antigen on non-malignant cells, the bystander effect, and cancer type [85].

ADCs have fewer side effects than the cytotoxic agents used in conventional chemotherapy. However, these side effects are considerable and have been described in a clinical study on solid tumors, including gynecologic cancers. We have described the TRAEs reported in each clinical trial in this review. TRAEs similar to those reported in chemotherapy and specific TRAEs for ADCs in clinical trials were found. MIRV was approved by the FDA for patients with OC. The common TRAE associated with this agent was ocular toxicity. In the Phase II trial of MIRV for OC, the overall incidence of TRAEs was 41% for blurred vision, 29% for keratopathy, 25% for dry eyes, and 13% for photophobia, while those in severe cases (over grade 3) were 6%, 9%, and 2%, respectively [45]. In the Phase II trial of TV for CC ocular toxicity was a TRAE. This trial reported conjunctivitis (26%) and dry eyes (23%) but no events over grade 3 occurred [39]. 

The reasons underlying these ocular adverse events are different. Those by TV are thought to have an on-target effect related to tissue factor expression in the conjunctiva. Thus, when administering TV, the use of vasoconstrictor eye drops and cooling eye pads are recommended. Ocular toxicity due to MIRV is an off-target effect caused by the DM4 payload. To prevent these events, using corticosteroids and lubricating eye drops, scheduling regular ophthalmic check-ups, and maintaining clear communication with ophthalmologists regarding new or worsening ocular symptoms, are recommended [86,87]. 

Zhu et al. reported pneumonitis (pneumonia) as the most common and fatal TRAE associated with ADCs, accounting for approximately 20% of deaths, based on 169 clinical trials involving 22,492 patients [88]. The incidence rate of pneumonitis was 10% in the Phase II trial for MIRV against OC [45]. T-DXd may be an effective ADC against OC. The incidence rate of pneumonitis was 27% in a Phase II trial for T-DXd against UCS. Kumagai et al., in an in vivo study, showed that this phenomenon was due to target-independent T-DXd uptake by alveolar macrophages and the release of payload as a mechanism of off-target toxicity. T- DXd-related ILD/pneumonitis incidence and severity are dependent on T-DXd dose and administration frequency [89].

Multidisciplinary guidelines for diagnosing and managing T-DXd-related interstitial lung disease (ILD)/pneumonitis have been published. Although usually of a low grade, T-DXd-related interstitial lung disease (ILD)/pneumonitis can be fatal in some cases. Early diagnosis, effective management, and close monitoring are crucial. A team including medical oncologists, pulmonologists, and other specialists should proactively monitor and manage ILD in patients receiving T-DXd. Treatment interruption is necessary for grade 1, and resolution using systemic steroids must be achieved before resuming treatment [90].

## 6. The New Development of ADCs for OC

### 6.1. Potential of Combining ADCs with Other Agents to Treat OC

Wei et al. have mentioned in their review that chemotherapy and ADCs act synergistically by targeting different phases of the cell cycle or modulating the expression of surface antigens on tumor cells [91]. Many chemotherapeutic drugs act as DNA-damaging agents that inhibit tumor cells by targeting the S phase of the cell cycle and inducing G2/M arrest. Microtubule-disrupting payloads, which are a part of ADCs, also target the G2/M phase of the cell cycle. Thus, they can be effectively combined with the DNA-damaging chemotherapeutic agents [91]. In fact, several studies have reported the potential efficacy of combination therapies with MIRV and chemotherapeutic agents against recurrent OC. A Phase I study of MIRV and gemcitabine combination therapy was conducted, and 20 patients with FRα-positive solid tumors, including PRR OC, recurrent EC, and triple-negative breast cancer, were enrolled. Nine of the twenty patients showed PR as their optimal response, while three patients (15%) exhibited confirmed PR. This trial demonstrated the promising efficacy of the MIRV-gemcitabine combination against PRR OC at the recommended Phase II dose; however, this combination also caused frequent hematologic toxicities [92]. Moore et al. have reported the efficacy of a MIRV-carboplatin combination therapy for patients with FRα-positive PSR OC in a Phase II study. The ORR was 71% (95% CI: 44–90) [44]. Meanwhile, Gilbert et al. have reported a Phase Ib/II trial of a MIRV-Bev combination therapy for PRR OC with an ORR of 44% (N = 94; 5 CR and 36 PR). The most common TRAEs were blurred vision, diarrhea, and nausea [93]. As shown in Table 3, Phase II/III trials of combination therapies with MIRV and various chemotherapeutic agents for OC are ongoing. These combinations are being developed for both recurrent and primary OCs [94]. Furthermore, the MIRV-Bev combination is being developed as part of a maintenance treatment for PSR OC [95].

In CC, Chiba et. al. investigated the association between the expression of TROP2 and the tumor immune microenvironment, including PD-L1, CD3, and CD8, using IHC in CC, reported that expression of TROP2 in CC is associated with increased levels of intratumorally tumor-infiltrating lymphocytes, and indicated that the potential of TROP2 targeted therapy in combination with immune checkpoint inhibitors [96]. 

The TROP2-targeted ADC, Dato-Dxd, is currently being investigated in Phase II trials in combination with anticancer agents, including immunotherapies and chemotherapies, for various solid tumors such as recurrent OC [97]. This trial is the first to evaluate ADCs combined with immunotherapy for patients with OC. 

Nicolò et al. mentioned that ADCs exert immunomodulatory activity by interacting with cancer and immune cells, and the synergy between ADCs and immunotherapy can overcome treatment resistance [98]. They also described the tumor-specific adaptive immunity induced by ADCs, which increases the infiltration of T cells into the tumor microenvironment while immune-checkpoint inhibitors reinvigorate exhausted T cells, enhancing antitumor immune responses [98]. Although the efficacy of immune-checkpoint inhibitor monotherapy or combination therapies for OC compared with conventional chemotherapies has not been reported [99,100], ADCs combined with immunotherapy may potentially benefit patients with gynecologic cancer, including OC, in the future.

### 6.2. Challenges for Resistance to ADCs

ADCs are a promising therapy class combining monoclonal antibodies with cytotoxic payloads. However, patients can develop resistance to ADCs. This resistance could be “antigen-related resistance” and “processing and payload resistance” [85,101,102]. This review is focused on “antigen-related resistance”. OC is heterogeneous in nature, with significant differences in incidence, behavior, and outcomes across histological subtypes [3].

Antigen-related resistance to ADCs in breast cancer has been reported. In an in vivo study, Loganzo et al. reported that reduced HER2 protein levels following months of treatment with HER2-ADCs may lead to HER2-ADC refractory cells and drug resistance [103]. Tumor heterogeneity in antigen expression can affect the efficacy of ADCs, and this has been observed in the KRISTINE trial, a Phase II study for Trastuzumab emtansine, a HER2-targeted ADC, plus pertuzumab in the neoadjuvant setting for patients with breast cancer. Notably, the primary endpoint was not met in any of the ten patients exhibiting tumor heterogeneity, resulting in a pathologic complete response (pCR) rate of 0%. Patients with high HER2 heterogeneity before treatment had worse PFS and OS compared with those showing low heterogeneity [104].

The bystander effect may contribute to solving this problem, whereby a drug released in cancer cells permeates the cell membrane and shows efficacy against surrounding cancer cells [105]. This bystander effect may be associated with the efficacy of T-DXd in UCS, a heterogeneous tumor, regardless of the HER2 expression status [48].

Recently, Saito et al. reported the wide expression of FRα in ovarian carcinosarcoma and UCS. High expression of FRα has been detected in 34% of HER2-negative uterine carcinosarcomas [106]. The effect of T-DXd on patients with HER2-positive recurrent UCS has been revealed in a Phase II trial [48]. Patients with HER2-negative recurrent UCS were not enrolled in this trial; therefore, no possible benefits of this agent have been reported. FRα is considered an attractive therapeutic target for such patients in the current context of HER2-targeted therapy [106]. Although these reports concern patients with ovarian carcinosarcoma and UCS, we believe that the findings apply to those with other cancers, including OC. ADCs have the potential to be considered as a complementary therapeutic option for tumors expressing multiple targets. As described earlier, the effects of several targeted ADCs for patients with OC have been reported in Phase II/III trials; clinical trials for other targeted ADCs are also underway. Thus, developing complementary therapeutic targets for ADCs is a possibility in the near future. This may contribute to solving antigen-related resistance.

Technologies have been developed to achieve higher efficacy of ADCs. Site-specific conjugation (SSC) is a method used to attach cytotoxic drugs to specific sites on antibody molecules in ADCs. These ADCs combine the highly specific targeting capabilities of antibodies with the delivery of a cytotoxic payload to specific cell types. SSC ensures that the cytotoxic drug is attached to the antibody at a specific site, enhancing the stability and efficacy of the ADCs while minimizing off-target effects. These methods can increase the therapeutic window of ADCs by improving their pharmacokinetic and pharmacodynamic properties [107]. One of the SSC’s tag-free enzymatic modification methods, which include the transglutaminase approach, the glycan remodeling approach, and the lipoic acid ligase approach, are focused on reasonable options by considering substrate selectivity. This technology has been expected as a potential tool to support drug developers as they go after antibody-conjugation approaches [108].

Advancements in payload technology have gained research attention. Payloads that target different cellular mechanisms, such as RNA polymerase inhibitors and immunomodulatory agents [109], have been introduced. HDP-101, an RNA polymerase II ADC targeting anti–B cell maturation antigen with an amanitin derivative, inhibited tumor growth in proliferating and resting multiple myeloma BCMA-positive cells in a preclinical study [110]. These new technologies for ADCs may contribute to overcoming resistance mechanisms and enhancing the efficacy of ADCs in treating various cancers, including OC.

## 7. Conclusions

OC is the most fatal of all gynecologic cancers. Although systematic treatments for OC, including platinum-based chemotherapy with Bev and PARPi, have been developed, the needs of patients resistant to these therapies have not been met. Recently, immunotherapy with immune checkpoint inhibitors has been used for treating solid tumors such as OC.

Unfortunately, no reports on the feasibility of this treatment over conventional therapies for patients with OC have been published.

ADCs, consisting of an antibody that targets tumor cells, a linker, and a payload, are novel therapeutic agents with no prior equivalent. If the target is expressed in tumors, the payload will reach them. This concept is especially favorable for treating OC due to its heterogeneous nature, with significant differences in incidence, behavior, and outcomes among histological subtypes [3].

MIRV, approved by the FDA, has shown efficacy against FRα-positive PRR OC compared to conventional monotherapy. Numerous advanced-phase clinical trials of ADCs against OC are underway. In this review, we summarized the development of ADCs for OC, as shown in Table 4. New ADCs, including combinations with chemotherapy, Bev, PARPi, and immunotherapy, are expected to benefit OC patients.

In conclusion, this review highlights the potential of ADCs for treating OC. Summarizing current evidence and trials, it emphasizes the efficacy of ADCs and discusses the new developments to address OC heterogeneity and resistance.

## Figures and Tables

**Figure 1 cancers-16-02545-f001:**
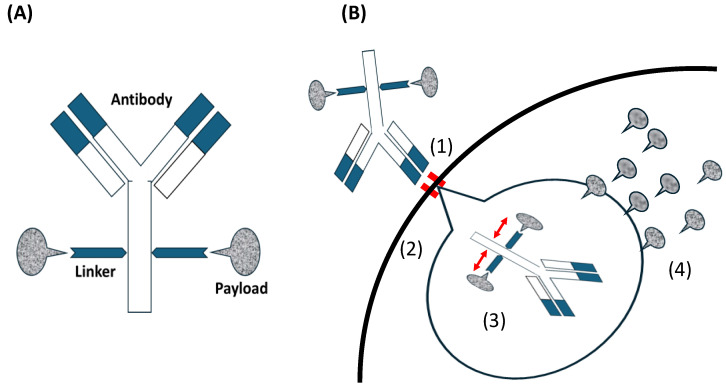
(**A**) Antibody-drug conjugates (ADCs) consist of an antibody, a linker, and a payload. (**B**) The mechanism of ADC action for inhibiting tumor cells; (1) the antibody in an ADC combines with an antigen as the ADC-receptor complex on the tumor. ‖: showing combined antibody with antigen. (2) ADC is internalized via antigen-mediated through receptor-mediated endocytosis and (3) cleaved into antibody and payload through endolysosomal processing. 
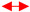
: showing released payload from linker. (4) The payload in a bioactive form that is released into the cytoplasm. Payloads that disrupt microtubule bind to tubulin. DNA-targeting payloads diffuse from the cytoplasm into the nucleus. Intracellular accumulation of the active payload results in cell death.

**Table 1 cancers-16-02545-t001:** Phase II/III trials of ADCs in Gynecologic cancer.

Author (Year)	Disease/ Including Status	Agent/ Target/ Paylord	Phase/ PE	Treatment Arm	PFS	OS	ORR	TRAEs
Moore et al. [42] (2021)	OC PRR and FRα- expression	MIRV/ FRα/ DM4	III PFS	ICC ^a^ vs MIRV	4.4 vs. 4.1, *p* = 0.897 (3.3 vs. 4.8, *p* = 0.049 ^b^)	*p* = 0.276 ^c^ (12.0 vs. 17.3, *p* = 0.063 ^d^)	10% vs. 24%, *p* = 0.014	MIRV showed a differentiated and more manageable safety profile than chemotherapy.
Moore et al. [43] (2023)	OC PRR, HGS and high FRα- expression	MIRV FRα/ DM4	III PFS	ICC ^a^ vs MIRV	3.98 vs. 5.62, *p* < 0.001	12.75 vs. 16.46, *p* = 0.005	15.9% vs. 42.3% *p* < 0.001	93.7% (over grade3: 54.1%) vs. 96.7% (over grade3: 41.7%) In MIRV group, grade 3 ocular adverse events of blurred vision occurred in 7.8%, keratopathy in 9.2%, and dry eye in 3.2%.
Moore et al. [44] (2022) #	OC PSR and FRα- expression	MIRV/ FRα/ DM4	II ORR	CP + MIRV	16.4 (95% CI, 10.4–30.2)	none data	71% (95% CI, 44–90)	The most common TRAEs (all grade, over grade 3): nausea (72%, no), diarrhea (67%, 6%), blurred vision (67%, 0%), thrombocytopenia (61%, 17%), fatigue (61%, 11%), and neutropenia (56%, 28%).
Matulonis et al. [45] (2023)	OC PRR, HGS and high FRα- expression	MIRV/ FRα/ DM4	II ORR	MIRV	4.3 (95% CI, 3.7–5.2)	13.8 (95% CI, 12.0–NR)	32.4% (95% CI, 23.6–42.2)	The most common TRAEs (all grade, over grade 3): blurred vision (41%, 6%), keratopathy (29%, 9%) and nausea (29%, no).
Banerjee et al. [46] (2018)	OC PRR	LIFA/ NaPi2b/MMAE	II PFS	PLD vs. LIFA	3.1 vs. 5.3 HR:0.78 (95% CI, 0.46–1.31 )	-	15% vs. 34% *p* = 0.03	Neuropathy, abdominal pain, diarrhea and neutropenia were significantly increased in LIFA.
Meric-Bernstam et al. [41] (2024)	OC+ PRR ***	T-Dxd/ HER2/ Deruxtecan	II ORR	T-Dxd	5.9 (95% CI, 4.0–8.3)	13.2 (95%CI, 8.0–17.7)	45.0% (95% CI, 29.3–61.5)	+
Lheureux et al. [47] (2022) #	OCC PRR and HGS or HGEM	AR/ Mesthelin/ Tubulinpo- lymerization inhibitor	II PFS	PB vs. ARB	9.6 vs. 5.3 HR:1.7 (95% CI, 0.9–3.4)	-	55% vs. 18%	The most common TRAEs in the ARB arm were mostly grade 1/2 increase AST (71%) and ALT (64%), thrombocytopenia (61%), fatigue (57%), and peripheral neuropathy (46%).
Vergote et al. [40] (2023) #	CC Recurrent or metastatic disease *	TV TF/ MMAE	III OS	ICC ^e^ vs TV	HR: 0.67 ^c^ (95% CI, 0.54–0.82)	9.5 vs. 11.5, HR:0.70 (95% CI:0.54–0.89)	5.2% vs. 17.8% *p* < 0.0001	85.4% (over grade 3: 45.2%) vs. 87.6% (over grade 3: 29.2%). AEs were consistent with the known TV safety profile, including for ocular, peripheral neuropathy, and bleeding AEs.
Coleman et al. [39] (2021)	CC Recurrent or metastatic disease *	TV TF/ MMAE	II ORR	TV	4.2 (95% CI, 3.0–4.4)	12.1 (95% CI, 9.6–13.9)	24% (95% CI, 16–33)	The most common TRAEs: conjunctivitis (26%), dry eye (23%), and keratitis (11%). None of serious TRAEs.
Meric-Bernstam et al. [41] (2024)	CC+ Recurrent or metastatic disease ***	T-Dxd/ HER2/ Deruxtecan	II ORR	T-Dxd	l 7.0 (95% CI, 4.2–11.1)	13.6 (95% CI, 11.1–NR)	50.0% (95% CI, 33.8–66.2)	+
Nishikawa and Hasegawa et al. [48] (2023)	UCS Recurrent with HER2 expression (≥1+) **	T-Dxd/ HER2/ Deruxtecan	II ORR	T-Dxd	6.7 (95% CI, 5.4–8.8)	15.8 (95% CI, 10.5–NR)	54.5% (95% CI, 32.2–75.6) ^f^ 70.0% (95% CI, 34.8–93.3) ^g^	Over grade 3 of TRAEs occurred in 61% of all, with the most common being decreased neutrophil count (27%), anemia (24%), and decreased lymphocyte count (21%).
Santin et al. [49] (2023) #	EC PRR, HGS and high FRα expression	IMMU-132 (SG)/ TROP2/Govitecan	II ORR	IMMU-132 ^h^	5.7	22.5	35%	The treatment was well-tolerated with no new or unexpected safety signals reported.
Meric-Bernstam et al. [41] (2024)	EC+ Locally advanced, metastatic disease or recurrence ***	T-Dxd/ HER2/ Deruxtecan	II ORR	T-Dxd	11.1 (95% CI, 7.1–NR)	26.0 (95% CI, 12.8–NR)	57.5% (95% CI, 40.9–73.0)	+Over grade 3 occurred in 40.8% of all, with the most common being neutropenia (10.9%) and anemia (10.9%). Serious event occurred in 13.5% patients. TRAEs resulting in death occurred in 1.5%.

^a^: Paclitaxel or Pegylated liposomal doxorubicin or Topotecan, ^b^: High FRα expression group, and based on the Hochberg procedure used in the statistical analysis plan for the study, this *p* value did not meet statistical significance; since the *p* value for the ITT was >0.05, this value was required to be <0.025 to be significant,. ^c^: High FRα expression group, There were no descriptions of median value in each, and had been described only *p* value., ^d^: High FRα expression group, ^e^: Topotecan or Vinorelbine or Gemcitabine or Irinotecan, ^f^: High HER2 expression, ^g^: Low HER2 expression, ^h^: Sacituzumab govitecan. #: Conference abstract, -: no data, +: Basket trial which are included any other solid tumors. TRAEs were described in patients with all tumors. *: Received two or fewer previous systemic regimens,**: With HER2 expression, and after systemic treatment (at least 1 regimen) or without alternative treatments, ***: Previously treated with chemotherapy, AEs: Adverse events, CC: Cervical cancer, CI: Confidence interval, EC: Endometrial cancer, EP: Endpoint, FRα:folate receptor alpha, HER2: Human epidermal growth factor receptor 2, HGS: High grade serous, ICC: Investigator‘s choice chemotherapy, MIRV: Mirvetuximab soravtansine, MMAE: monomethyl auristatin E, NR: Not reported, OC: Ovarian cancer, ORR: Overall response rate, OS: Overall survival, PE: Primary endpoint, PFS: Progression free survival, PRR: Platinum-resistant recurrent, PSR: Platinum-sensitive recurrent, SG: Sacituzumab govitecan, T-Dxd: Trastuzumab deruxtecan, TF: Tissue factor, TROP2: Trophoblast cell surface antigen 2, TRAEs: Treatment-related adverse events, TV: Tisotumab vedotin, UCS: Uterine carcinosarcoma.

**Table 2 cancers-16-02545-t002:** Ongoing Phase II/III trials of ADCs monotherapy in Ovarian cancer.

Primary or Recurrent	Including Status	Agent	Target	Paylord	ClinicalTrials.gov Identifier (Trial Name)	Phase	RCT	Description
Recurrent	PRR and High grade carcinoma	DS-6000	CDH6	Deruxtecan	NCT06161025	II/III	Yes	ICC ^a^ vs. DS-6000
Recurrent	PSR, HGSC and NaPi2b-positive	UpRi	NaPi2b	AF-HPA	NCT05329545	III	Yes	After PR or CR of platinum-based therapy, UpRi vs. Placebo
Recurrent	PRR	IMGN853 (MIRV)	FRα	DM4	NCT05622890	III	No	IMGN853 (MIRV)
Recurrent	PRR and HGSC	MORAb-202	FRα	Ecteribulin	NCT05613088	II	Yes	ICC ^b^ vs. MORAb-202
Recurrent	PRR	IMGN853 (MIRV)	FRα	DM4	NCT05622890	II	No	IMGN853
Recurrent	PSR	MIRV	FRα	DM4	NCT05887609	II	Yes	After chemotherapy including platinum drug as maintenance Olaparib vs. MIRV
Recurrent	PRR	STRO-002 (luvelta)	FRα	SC209	NCT06238687	II	No	STRO-002
Recurrent	PRR	Dato-DXd	TROP2	Deruxtecan	NCT05489211	II	No	Dato-DXD
Recurrent	PRR	ESG401	TROP2	SN-38	NCT04892342	II	No	ESG401
Recurrent	PRR	IMMU-132 (Sacituzumab Govitecan)	TROP2	Govitecan	NCT06028932	II	No	IMMU-132
Recurrent	PRR	SHR-A1811 (Trastuzumab rezetecan)	HER2	SHR9265 (topoisomerase I inhibitor)	NCT05896020	II	No	SHR- A1811
Recurrent	PRR	RC48 (Disitamab vedotin)	HER2	MMAE	NCT06003231	II	No	RC48
Recurrent	PRR	BL-B01D1	EGFR xHER3	Ed-04	NCT05803018	II	No	BL-B01D1
Recurrent	PRR	HS-20089	B7-H4	Undisclosed Payload	NCT06014190	II	No	HS-20089
Recurrent	PRR	BA3021	Ror2	Vedotin	NCT04918186	II	No	BA3021

^a^: Gemcitabine or Paclitaxel or Topotecan or Pegylated liposomal doxorubicin, ^b^: Paclitaxel or Pegylated liposomal doxorubicin or Topotecan. AEs: Adverse events, AF-HPA: Auristatin F-hydroxypropylamide, CDH6: Cadherin-6, CI: Confidence interval, FRα:folate receptor alpha, HER2: Human epidermal growth factor receptor 2, HER3: Human epidermal growth factor receptor 3, HGS: High-grade serous, ICC: Investigator‘s choice chemotherapy, luvelata: Luveltamab tazevibulin, MIRV: Mirvetuximab soravtansine, MMAE: monomethyl auristatin E, NR: Not reported, OC: Ovarian cancer, ORR: Overall response rate, OS: Overall survival, PFS: Progression free survival, PRR: Platinum-resistant recurrent, PSR: Platinum-sensitive recurrent, SG: Sacituzumab govitecan, T-Dxd: Trastuzumab deruxtecan, TF: Tissue factor, TROP2: Trophoblast cell surface antigen 2, TRAEs: Treatment-related adverse events, TV: Tisotumab vedotin, UCS: Uterine carcinosarcoma, UpRi: Upifitamab Rilsodotin.

**Table 3 cancers-16-02545-t003:** Ongoing Phase II/III trials of ADC combination therapy in OC.

Primary or Recurrent	Including Status	Agent	Target	Paylord	ClinicalTrials.gov Identifier (Trial Name)	Phase	RCT	Description
Recurrent	PSR and high FRα expression	MIRV	FRα	DM4	NCT05445778 (GLORIOSA)	III	Yes	Platinum-based therapy + Bev and Bev maintenance, vs. Platinum-based therapy + Bev and Bev + MIRV maintenance
Primary	Newly diagnosed, advanced-stage HGSC	MIRV	FRα	DM4	NCT04606914	II	No	CP + MIRV
Recurrent	PSR	MIRV	FRα	DM4	NCT05887609	II	Yes	After chemotherapy including platinum drug as maintenance Olaparib vs. MIRV
Recurrent	PSR	AZD5335	FRα	AZ14170132 (Topo I inhibitor)	NCT05797168 (FONTANA)	II	No	Saruparib + AZD5335
Recurrent	PSR or PRR	Dato-DXd	TROP2	Deruxtecan	NCT05489211 (TROPION-PanTumor03)	II	No	CP + Dato-DXD → Saruparib + Dato-DXD (PSR) Dato-DXD monotherapy (PRR)

AEs: Adverse events, Bev: Bevacizumab, CDH6: Cadherin-6, CI: Confidence interval, CP: Carboplatin, Dato-DXd: Datopotamab deruxtecan, FRα:folate receptor alpha, HER2: Human epidermal growth factor receptor 2, HER3: Human epidermal growth factor receptor 3, HGS: High grade serous, ICC: Investigator‘s choice chemotherapy, luvelata: Luveltamab tazevibulin, MIRV: Mirvetuximab soravtansine, MMAE: monomethyl auristatin E, NR: Not reported, OC: Ovarian cancer, ORR: Overall response rate, OS: Overall survival, PFS: Progression free survival, PRR: Platinum-resistant recurrent, PSR: Platinum-sensitive recurrent, SG: Sacituzumab govitecan, T-Dxd: Trastuzumab deruxtecan, TF: Tissue factor, TROP2: Trophoblast cell surface antigen 2, TRAEs: Treatment-related adverse events, TV: Tisotumab vedotin, UCS: Uterine carcinosarcoma.

**Table 4 cancers-16-02545-t004:** Summarized the development of ADCs for Ovarian cancer.

Target Antigen (Expression% in OC)	Agent Name	Anti- Body Type	Linker Name (Type)	Paylord Name (Target)	Common TRAEs	Development Status for OC	Development of Combination Therapy
FRα (50–80%)	MIRV	IgG1-kappa	N-Succinimidyl 4-(2 pyridyldithio)-2-sulfobutanoate linker (Cleavable)	DM4 (tubulin)	ocular events, diarrhea, fatigue, nausea, vomiting, peripheral, neuropathy, netropenia	FDA approved	O
	MORAb-202	IgG1-kappa	A reduced interchain disulfide bonds to maleimido-PEG2-valine-citrulline-p-aminobenzylcarbamyl linker (Cleavable)	Eribulin (tubulin)	ILD/pneumonitis, nausea, pyrexia, malaise, headache	ongoing Phase II	
	STRO-002 (luvelta)	IgG1	valine citrulline p-aminobenzyl carbamate linker (Cleavable)	SC209 (tubulin)	neutropenia, arthralgia, anemia, neutropenia	ongoing Phase II	
TROP2 (50–60%)	Dato-DXd	IgG1	A tetrapeptide-based linker (Cleavable)	Deruxtecan (topoisomerase I)	nausea, anemia, decreased WBC, ILD/pneumonitis,	ongoing Phase II	O
	ESG401	IgG1	unrevealed linker	SN38 (topoisomerase I)	leukopenia, neutropenia, anemia, fatigue, nausea, vomiting, thrombocytop-enia, diarrhea, skin rash, oral mucositis	ongoing Phase II	
	IMMU-132 (SG)	IgG1-kappa	hRS7 via a hydrolysable CL2A linker (cleavable)	Govitecan (topoisomerase I)	neutropenia, decreased WBC, anaemia, diarrhoea, fatigue, febrile, neutropenia, hypophosphatemia, diarrhoea	ongoing Phase II	
HER2 (12–30%)	T-DXd	IgG1-kappa	Gly-Phe-Leu-Gly (tetrapeptide)	Deruxtecan (topoisomerase I)	nausea, anemia, diarrhea, vomiting, fatigue, neutropenia, ILD/pneumonitis	possibility FDA approved *	
	SHR-A1811 (Trastumab rezetecan)	IgG1-kappa	unrevealed (cleavable)	SHR9265 (topoisomerase I)	neutropenia, anemia, decreased WBC, ILD/pneumonitis	ongoing Phase II	
	RC48 (Disitamab vedotin)	IgG1-kappa	mc-val-cit-PABC (cleavable)	MMAE (tubulin)	peripheral sensory neuropathy, leukopenia, neutropenia, AST/ALT increased, alopecia, asthenia, decreased appetite	ongoing Phase II	
NaPi2b (66%)	UpRi	IgG1-kappa	Fleximer polymer scaffold (cleavable)	AF-HPA (tubulin)	AST increased, fatigue, anemia, thrombocytop-enia, neutropenia, peripheral neuropathy, ocular toxicity, ILD/pneumonitis	ongoing Phase III	
CDH6 (85%)	DS-6000 (R-Dxd)	IgG1	Tetrapeptide based linker (cleavable)	Deruxtecan (topoisomerase I)	nausea, fatigue, vomiting, diarrhoea **	ongoing Phase II/III	

*: The FDA will give accelerated approval for T-DXd to treat any advanced solid cancer that expresses HER2. **: TRAEs were reported as the most common over grade 3 due to a report by only conference. AF-HPA: Auristatin F-hydroxypropylamide, CDH6: Cadherin-6, FRα: folate receptor alpha, HER2: Human epidermal growth factor receptor 2, IgG: immunoglobulin G, ILD: interstitial lung disease, luvelata: Luveltamab tazevibulin, MIRV: Mirvetuximab soravtansine, MMAE: monomethyl auristatin E, OC: Ovarian cancer, SG: Sacituzumab govitecan, T-Dxd: Trastuzumab deruxtecan, TROP2: Trophoblast cell surface antigen 2, TRAEs: Treatment-related adverse events, UpRi: Upifitamab Rilsodotin.

## Data Availability

Not applicable.

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
