# Peer review of "Antibody-Drug Conjugates: The New Treatment Approaches for Ovarian Cancer"

_cancers, 2024, doi:10.3390/cancers16142545_

Round 1

Reviewer 1 Report

Comments and Suggestions for Authors

This manuscript reviews the latest finding and ongoing clinical trials on antibody drug conjugates (ADC) used to treat gynaecologic cancer and ovarian cancer. The authors discussed several key points including the mechanisms of actions, the efficacy, the side effects of the currently approved ADCs as well as the opportunity to use this new class of drugs in combination therapy with chemotherapy or immunotherapy.

The manuscript contains lot of numbers and information which is very pleasant. However, it would be good to improve the organization of the information. There are several repetitions due to a lack hierarchy of the information. It may be better to reorganize the manuscript as follow: Introduction, mechanisms of action, efficacy of ADCs, side/advent effect of the treatment, the future therapeutic strategy and finally the conclusion... or something like that.

It would be very informative to include a table that list the different components of ADCs used today (approved, rejected or clinical trials). Thus, for each ADC, list in different columns name, tumour cell surface antigen (and its frequence in OC tumour cells), antibody type (info on IgG), the linker (sequence and crosslinking moiety), the payload name as well as its target (microtubule or DNA /proliferating or non-proliferating cells) and eventually the most likely non-tumour/normal cells to be affected by this ADC. Alternatively, a graphical scheme with this information is also ok. This would improve a lot the manuscript and provide the readers with important summary of information and would help in the development of new ADCs.

Comments on the Quality of English Language

The authors should be consistent in the terms as they use lot of abbreviations. For example, sometime the side effects are termed TRAEs whereas some time it is termed TEAEs. I guess this may typos... Also, there are lot of grammatical errors…Thus, please read carefully all the manuscript and correct all the language errors.

Reviewer 2 Report

Comments and Suggestions for Authors

This masucript presents a comprehensive review of Antibody-Drug Conjugates (ADCs) as innovative therapeutic agents for ovarian cancer (OC). The authors have detailed the mechanisms, clinical efficacy, and adverse events associated with ADCs, providing valuable insights into their potential in OC treatment. However, several areas could be improved to enhance the manuscript's clarity, depth, and impact.

Major Comments:

1. Clarity and Focus:

   - Specificity of ADC Targets: While the manuscript describes the general mechanism of ADCs, it could benefit from a more detailed discussion on the specificity of ADC targets. Highlighting the rationale for selecting specific targets in ovarian cancer, such as FRα, TROP2, and HER2, would provide a clearer understanding of their importance.

   - Comparison with Existing Therapies: A comparative analysis of ADCs with existing OC therapies, including conventional chemotherapy and targeted therapies, would strengthen the manuscript. Discussing the advantages and limitations of ADCs relative to these therapies could offer a more balanced perspective.

2. Critical Analysis of Clinical Trials:

   - Detailed Outcomes: The manuscript includes data from several clinical trials but lacks a critical analysis of the results. Discussing the statistical significance, study limitations, and potential biases in these trials would provide a more nuanced view of the efficacy and safety of ADCs.

   - Adverse Events Management: While the manuscript mentions treatment-related adverse events (TRAEs), it should elaborate on the management strategies for these events. Providing recommendations for mitigating TRAEs, such as dose adjustments or supportive care measures, would be valuable for clinicians.

3. Mechanistic Insights:

   - Resistance Mechanisms: The manuscript could be improved by discussing potential resistance mechanisms to ADCs in ovarian cancer. Exploring how tumor heterogeneity and genetic mutations might influence ADC efficacy and resistance could provide deeper insights into the challenges of ADC therapy.

   - Innovations in ADC Design: This section could be further elaborated to include the latest advancements in ADC technology, particularly focusing on site-specific conjugation (SSC) and novel payloads.

     - Site-Specific Conjugation (SSC): 

       The manuscript should explore the benefits of SSC in improving the therapeutic index of ADCs. SSC techniques ensure that the cytotoxic drug is attached to the antibody at a specific site, enhancing the stability and efficacy of the ADC while minimizing off-target effects. The authors could discuss the chemical site-specific modification methods and tag-free enzymatic modification methods that have been highlighted in recent reviews. These methods have shown promise in increasing the therapeutic window of ADCs by improving their pharmacokinetic and pharmacodynamic properties.

     - Novel Payloads:

       The manuscript could also delve into the potential of novel payloads beyond traditional cytotoxic agents. Recent advancements have introduced payloads that target different cellular mechanisms, such as RNA polymerase inhibitors and immunomodulatory agents. These new payloads offer opportunities to overcome resistance mechanisms and enhance the efficacy of ADCs in treating various cancers, including ovarian cancer. See and cite this comprehensive review for this discussion.

   https://www.tandfonline.com/doi/abs/10.1080/14712598.2023.2276873

4. Future Directions and Research Gaps:

   - Combination Therapies: The manuscript briefly mentions combination therapies with ADCs. A more detailed exploration of ongoing and future research into combination strategies, including synergistic effects with immunotherapies or other targeted agents, would enhance the manuscript's forward-looking perspective.

   - Personalized Medicine: Discussing the role of biomarkers and personalized medicine in optimizing ADC therapy for ovarian cancer patients would be a valuable addition. Addressing how patient selection based on molecular profiling can improve treatment outcomes would align with current trends in oncology.

 Minor Comments:

Figures and Tables: While the figures and tables are informative, adding more visual aids, such as flowcharts depicting the ADC mechanism or graphs comparing clinical trial outcomes, would enhance readability and comprehension.

Reviewer 3 Report

Comments and Suggestions for Authors

Sato et al. provide a review of antibody-drug conjugates (ADCs) in the context of ovarian cancer. While the subject matter is compelling and holds significant clinical relevance, several areas warrant further elaboration and refinement:

1. The authors should provide a comprehensive discussion on the structure of ADCs, elucidating how various components, such as the cytotoxic payload and linker, contribute to their functional properties.

2. The figure legends require additional mechanistic details to enhance clarity and understanding.

3. It would be more beneficial for the review to incorporate findings from the latest primary research articles rather than relying heavily on information from other reviews, such as the one by Zhu et al.

4. The manuscript should include more recent references to reflect the current state of research in this field.

5. Future directions for the development and application of ADCs in ovarian cancer should be discussed to provide a forward-looking perspective.

6. The authors should acknowledge FDA-approved ADCs for other types of cancer to contextualize the progress and potential of ADCs in ovarian cancer.

Comments on the Quality of English Language

Can be improved.

Reviewer 4 Report

Comments and Suggestions for Authors

The aim of this review was to highlight the role of Antibody-Drug Conjugates (ADC) in the treatment of Ovarian Cancer.

This innovative manuscript is well written; language, grammar and punctuation are right and technical terms are spelt correctly. Authors should be congratulated on original research.

I would also like to draw attention of authors to the fact that there are several points in this manuscript.

Please find below my consideration:

-       Abstract: Page 1, Line 26: authors should specify the acronym PRR: platinum resistant recurrence;

-       Keywords: Page 1, Line 39: authors should increase the number of keywords and explain the acronym ADCs (Antibody-Drug Conjugates)

-       Introduction: Page 1, Lines 42-43: authors should use a more recent revision than number “1” to define the incidence and mortality of ovarian cancer

-       Treatment-related adverse events with ADCs: Page 3 Lines 128-131: authors could better explain the reasons of the onset of ocular adverse events

-       Treatment-related adverse events with ADCs: Page 4, Lines 136-141: authors should provide summary details on the treatment of pneumonias induced by ADCs. (Swain SM, Nishino M, Lancaster LH, Li BT, Nicholson AG, Bartholmai BJ, Naidoo J, Schumacher-Wulf E, Shitara K, Tsurutani J, Conte P, Kato T, Andre F, Powell CA. Multidisciplinary clinical guidance on trastuzumab deruxtecan (T-DXd)-related interstitial lung disease/pneumonitis-Focus on proactive monitoring, diagnosis, and management. Cancer Treat Rev. 2022 May;106:102378. doi: 10.1016/j.ctrv.2022.102378.)

-       Table 1, Page 5-7; Table 2 Page 9 and Table 3 Page 11: authors should improve the graphics and structure of the table (divide rows and columns)

Round 2

Reviewer 2 Report

Comments and Suggestions for Authors

The authors have appropriately addressed the revisions.

However, in the context of SSC, the reviewers consider tag-free enzymatic modification methods to be significant and recommend adding appropriate citations

Additionally, Reference 106 is outdated and focused primarily on research aspects. It is suggested to replace it with a more practical and recent review.
